# Design and Synthesis of Novel Breast Cancer Therapeutic Drug Candidates Based upon the Hydrophobic Feedback Approach of Antiestrogens

**DOI:** 10.3390/molecules24213966

**Published:** 2019-11-01

**Authors:** Kiminori Ohta, Asako Kaise, Fumi Taguchi, Sayaka Aoto, Takumi Ogawa, Yasuyuki Endo

**Affiliations:** 1School of Pharmacy, Showa University, 1-5-8, Hatanodai, Shinagawa-ku, Tokyo 142-8555, Japan; 2Faculty of Pharmaceutical Sciences, Tohoku Medical and Pharmaceutical University, 4-4-1 Komatsushima, Aoba-ku, Sendai 981-8558, Japan; kaise@tohoku-mpu.ac.jp (A.K.); f0u1m2i9@docomo.ne.jp (F.T.); syk-at1213@docomo.ne.jp (S.A.); ogawa-takumi@pmda.go.jp (T.O.)

**Keywords:** sulfur, anticancer, estrogen

## Abstract

Based upon hydrophobic feedback approaches, we designed and synthesized novel sulfur-containing ERα modulators (**4** and **5**) as breast cancer therapeutic drug candidates. The tetrahydrothiepine derivative **5a** showed the highest binding affinity toward ERα because of its high hydrophobicity, and it acted as an agonist toward MCF-7 cell proliferation. The corresponding alkylamino derivative **5d** maintained high binding affinity to ERα and potently inhibited MCF-7 cell proliferation (IC_50_: 0.09 μM). Docking simulation studies of compound **5d** with the ERα BD revealed that the large hydrophobic moiety of compound **5d** fit well into the hydrophobic pocket of the ERα LBD and that the sulfur atom of compound **5d** formed a sulfur–π interaction with the amino acid residue His524 of the ERα LBD. These interactions play important roles for the binding affinity of compound **5d** to the ERα LBD.

## 1. Introduction

17β-Estradiol (E2) is an endogenous female hormone and plays an important role in the female reproductive system. Estrogens including E2 influence the growth, differentiation, and functioning of many target tissues mediated by the binding to estrogen receptor α (ERα) [1,2,3]. Although activation of ERα is essential for the maintenance of homeostasis of female functions, they can sometimes induce and cause progression of estrogen-dependent breast cancers [4,5,6]. Several antiestrogens have been developed to prevent and control hormone-responsive breast cancer [7,8,9,10]. Tamoxifen has been used worldwide for more than 40 years for the treatment of estrogen-dependent breast cancer (Figure 1) [11,12,13]. Interestingly, tamoxifen acts as either an agonist or an antagonist depending on tissue type; tamoxifen exhibits antiestrogenic action in breast cancer and hot flashes, and estrogenic action in bone and cholesterol metabolism, and is a selective estrogen receptor modulator (SERM) [11,12,13]. The dimethylaminoethyl chain of tamoxifen is attached to a benzene ring, where it plays a strategic role in the expression of antiestrogenic activity. The triphenylethylene moiety of tamoxifen plays an important role in controlling ERα binding affinity. The triphenylethylene moiety is a promising structure for the development of an ER ligand. The triphenylethylene structure exhibits geometric isomers *E* and *Z* that are caused by the key alkylamino chain, and the asymmetric hydrophobic part and the isomers easily isomerize between the *E* and *Z* forms. Therefore, the isomerization often become a big problem in the synthesis, purification, and preservation of an either isomer. Indeed, 4-hydroxytamoxifen, an active metabolite of tamoxifen, easily isomerizes from the active *Z* form to the inactive *E* form in solution and in in vitro studies [14,15,16,17]. In this regard, novel ER antagonists comprised of other skeleton structures, such as benzothiophene [18], dihydronaphthalene [19], benzopyrane [20], and steroid [21] structures, have been developed as ER antagonists.

On the other hand, we found that a simple 1,2-bis(4-hydroxyphenyl)-*o*-carborane, BE360 (**1**) exhibited high binding affinity toward ERα and estrogenic action in bone without showing estrogenic action in the uterus of ovariectomized (OVX) and orchidectomized (ORX) mice (Figure 1) [22]. That is, compound **1** acted as an agonist in bone and an antagonist in female reproductive tissues, despite lack of an alkylamino chain. Based upon the hydrophobic feedback approaches, we focused on three-dimensional hydrocarbon units as new hydrophobic pharmacophores to design and synthesize BE1060 (**2**) and BE1054 (**3**) (Figure 1) [23]. Although the tetramethylcyclohexene-based bisphenol compound **3** acted as a partial agonist, similar to compound **1**, the bicyclo[2,2,2]octene-based bisphenol compound **2** showed potent estrogenic activity. Unlike the triphenylethylene structure, a vicinal bisphenol structure does not have a geometric isomer, even with the key alkylamino chain. We were interested in the effects of ring size and sulfur atom in the hydrophobic structure on ER activity. Furthermore, to understand the effects of sulfur, sulfone, and sulfoxide on the biological activity, we designed dihydrothiophene and tetrahydrothiepine derivatives (**4a**–**4d** and **5a**–**5d**) as novel ER antagonist candidates. Herein, we describe the facile synthesis and biological activities of the designated compounds **4a**–**4d** and **5a**–**5d** (Figure 1).

## 2. Results and Discussion

### 2.1. Chemistry

Synthesis of the dihydrothiophene derivatives (**4a**–**4d**) is summarized in Scheme 1. Commercially available 2-bromo-4′-methoxyacetophenone (**6**) was reacted with sodium sulfide (Na_2_S) to form the sulfide compound **7** in 95% yield [24]. Intramolecular McMurry coupling of **7** resulted in formation of the dihydrothiophene derivative **8** in 60% yield, which was demethylated with BBr_3_ to form the target compound **4a** in 97% yield [25]. Although the dihydrothiophene derivative **8** was treated with 1 equimolar amount of *m*-chloroperbenzoic acid (*m*-CPBA) to form the sulfoxide compound **9** in 70% yield, followed by demethylation with BBr_3_ to form the decomposition product of compound **9**, it did not result in the formation of compound **4b**. Thus, we carried out oxidation of compound **4a** with *m*-CPBA to form the desired compound **4b** in 73% yield. When compound **8** was treated with two equimolar amounts of *m*-CPBA, the corresponding sulfone compound **10** was obtained and was then demethylated with BBr_3_ to form compound **4c** in quantitative yield over two steps. Mitsunobu reaction of compound **4a** with 3-dimethylamino-1-propanol resulted in formation of compound **4d** with a dimethylaminopropyl chain in 37% yield [26].

Scheme 2 summarizes the synthesis of tetrahydrothiepine derivatives (**5a**–**5d**). A starting material (**12**) for the seven-membered ring derivative **5** was synthesized by Friedel-Craft acylation of anisole with acyl chloride **11**, followed by sulfide formation with Na_2_S to form compound **13** in 84% yield over two steps [24]. The tetrahydrothiepine derivatives (**5a**–**5d**) were synthesized in the same manner as those of compounds **4a**–**4d**. Briefly, intramolecular McMurry coupling of compound **13**, followed by demethylation, resulted in formation of the sulfide compound **5a** in a 66% yield [25]. Compound **5a** was transformed into the sulfoxide compound **5b** by oxidation with one equimolar amount of *m*-CPBA in a 30% yield. Moreover, compound **5a** was reacted with 3-dimethylamino-1-propanol under Mitsunobu reaction conditions to form compound **5d** in a 28% yield [26]. Treatment of compound **14** with two equimolar amounts of *m*-CPBA followed by demethylation resulted in the formation of the sulfone compound **5c** in a 49% yield over two steps.

### 2.2. Biological Evaluation

The binding affinities of the synthesized bisphenols (compounds **4a**–**4c** and **5a**–**5c**) toward ERα were evaluated by means of a competitive binding assay using [6,7-^3^H] 17β-estradiol and human recombinant ERα [27,28,29]. Figure 2A,B shows dose-response curves for competitive binding of the synthesized compounds **4a**–**4c** and **5a**–**5c** to ERα, respectively. Compound **4a**, containing a sulfide group, moderately bound to ERα and the binding affinity was about 100 times lower than that of E2. The sulfoxide (**4b**) and sulfone (**4c**) compounds did not bind to the ERα ligand-binding domain (LBD), even at concentrations 1000 times higher than that of E2. The binding affinity of the tetrahydrothiepine derivative **5** toward ERα showed a similar tendency to that of compound **4**, and the sulfide derivative **5a** bound to ERα more potently than compound **4a**. The tamoxifen-inspired derivatives **4d** and **5d**, both containing a dimethylaminopropyl group, showed similar binding affinities to compounds **4a** and **5a**, respectively. With the binding affinity of E2 to ERα taken as 100, the relative binding affinity (RBA) values of compounds **4d** and **5d** were 4.0 and 37.1, respectively (Table 1). It is suggested that the binding affinities of compounds **4** and **5** depend on the hydrophobicity of the ring structures and the polarity of the sulfur atoms, which can be understood from the CLogP values of compounds **4a**–**4d** and **5a**–**5d** in Table 1.

Next, estrogenic and antiestrogenic activities of compounds **4a**–**4d** and **5a**–**5d**, except for **4b** and **5b** (poor binding affinity toward ERa), were evaluated by means of a cell proliferation assay using the human breast cancer cell line MCF-7, which shows ER-dependent growth [27,28,29]. Table 1 summarizes the EC_50_, IC_50_, and E_max_ values estimated from the MCF-7 cell proliferation assays. E_max_ represents the maximal efficacy of each compound toward cell proliferation and is an index of ER partial agonistic activity. E_max_ was estimated relative to that of E2, which was taken as 100. Although the binding affinity of compound **4a** was ~10 times higher than that of compound **4c**, both compounds showed estrogenic activity with similar EC_50_ values. Based on the E_max_ values of compounds **4a** and **4c**, we suggest that the complex formed between compound **4c** and the ERα LBD is more active for transcription than that formed between compound **4a** and the ERα LBD. Compound **5a**, which exhibited the most potent binding affinity to ERα LBD, also showed the most potent MCF-7 cell proliferation activity, with an EC_50_ value of 0.048 μM. As expected from the weak binding affinity to ERα LBD, compound **5c** showed weak estrogenic activity. On the other hand, the tamoxifen-inspired derivatives **4d** and **5d**, containing alkylamino chains, potently inhibited cell proliferation of MCF-7 cells induced by 0.1 nM of E2, with IC_50_ values of 0.41 and 0.09 μM, respectively.

### 2.3. Docking Study

To understand the docking modes of the sulfur-containing ligands **4d** and **5d** with the ERα LBD, docking simulation studies of both compounds toward the human ERα LBD (PDB: 1ERR) were carried out using an automatic docking program (Discovery Studio 2018/CDOCKER) [30]. The docking mode of compound **4d** was superimposed onto that of compound **5d** at the ERα LBD (Figure 3A,B). The docking mode of compound **4d** toward the ERα LBD was similar to that of compound **5d**. Each phenolic hydroxy group of compounds **4d** and **5d** formed hydrogen bonds with the amino acid residues Glu353 and Arg394 (Figure 3C), and each dimethylaminopropyl chain of compounds **4d** and **5d** was found to be located within the same space of the ERα LBD (Figure 3A). The hydrophobic moiety of compound **5d** was spread widely across the hydrophobic pocket of the ERα LBD and the sulfur atom of the tetrahydrothiepine ring formed a sulfur–π interaction with the amino acid residue His524 (Figure 3B,C). Docking scores (CDOCKER interaction energy) of compounds **4d** and **5d** toward the ERα LBD were 51.1 and 54.0, respectively, and the order of docking score was correlated with the results of the binding assay. Therefore, we attributed the potent binding affinity of compound **5d** toward the ERα LBD to both hydrophobic interactions and sulfur–π interactions.

## 3. Materials and Methods

### 3.1. General Consideration

Melting points were determined with an MP-J3 micro melting point apparatus (Yanaco, Kyoto, Japan) and were not corrected. ^1^H NMR and ^13^C NMR spectra were recorded with JNM-LA-400 spectrometer (JEOL, Tokyo, Japan). Chemical shifts for ^1^H NMR spectra were referenced to tetramethylsilane (0.0 ppm) as an internal standard. Chemical shifts for ^13^C NMR spectra were referenced to residual ^13^C present in deuterated solvents. The splitting patterns are designated as follows: s (singlet), d (doublet), t (triplet), q (quartet) and m (multiplet). Mass spectra were recorded on a JEOL JMS-DX-303 spectrometer (JEOL, Tokyo, Japan). Elemental analyses were performed with a Perkin Elmer 2400 CHN spectrometer (PerkinElmer, Winter Street Waltham, MA, USA). Column chromatography was carried out using Merck silica gel 60 (0.063–0.200 μm) and TLC was performed on Merck silica gel F254. 

### 3.2. Synthesis

*1-(4-Methoxyphenyl)-2-[2-(4-methoxyphenyl)-2-oxo-ethylsulfanyl] ethanone* (**7**). To a stirred solution of **6** (2.80 g, 12.2 mmol) in acetone (40 mL) was added a solution of Na_2_S 9H_2_O (1.50 g, 6.25 mmol) in H_2_O (12 mL) at 0 °C. After completion of the addition, the mixture was warmed to room temperature and stirred for 2.5 h. The solvents were removed, and the residue was recrystallized from MeOH to give **7** (1.91 g, 5.78 mmol, 95 %) as a colorless solid; ^1^H NMR (400 MHz, CDCl_3_) *δ* (ppm) 3.87 (6H, s), 3.93 (4H, s), 6.93 (4H, d, *J* = 8.8 Hz), 7.95 (4H, d, *J* = 8.8 Hz); ^13^C NMR (100 MHz, CDCl_3_) *δ* (ppm) 37.4, 55.5, 113.9, 128.4, 131.0, 163.8, 193.0; MS (EI) *m/z* = 330 (M^+^), 135 (100%).

*3,4-Bis-(4-methoxyphenyl)-2,5-dihydrothiophene* (**8**). To a suspension of Zn powder (1.57 g, 24.0 mmol) in dry THF (60 mL) was added TiCl_4_ (1.3 mL, 12.0 mmol) at −30 °C, and the mixture was refluxed for 2.5 h. A solution of **7** (1.0 g, 3.0 mmol) in dry THF (45mL) was slowly added to the reaction mixture at 0°C. The reaction mixture was refluxed for 2 h and then quenched with 10% K_2_CO_3_ aqueous solution. The insoluble materials were filtered through Celite, and the filtrate was extracted with AcOEt. The organic layer was washed with brine, dried over Na_2_SO_4_, and concentrated. Purification by silica gel column chromatography (eluent: AcOEt/*n*-hexane = 1/30) gave **8** as a colorless solid (0.55 g, 1.84 mmol, 61 %); ^1^H NMR (400 MHz, CDCl_3_) *δ* (ppm) 3.77 (6H, s), 4.22 (4H, s), 6.75 (4H, d, *J* = 8.8 Hz), 7.05 (4H, d, *J* = 8.8 Hz); ^13^C NMR (100 MHz, CDCl_3_) *δ* (ppm) 43.8, 55.1, 113.6, 129.4, 129.7, 134.3, 158.6; MS (EI) *m/z* = 298 (M^+^, 100%); Anal. Calcd for C_18_H_18_O_2_S: C, 72.45; H, 6.08, Found: C, 72.13; H, 6.20.

*3,4-Bis-(4-hydroxyphenyl)-2,5-dihydrothiophene* (**4a**). To a solution of **8** (407 mg, 1.36 mmol) in dry CH_2_Cl_2_ (9 mL) was added 1M of a solution of BBr_3_ in CH_2_Cl_2_ (3.4 mL, 3.4 mmol) at −80 °C, and the mixture was stirred at room temperature for 24 h. The mixture was poured onto ice and extracted with AcOEt. The organic layer was washed with brine, dried over MgSO_4_, and concentrated. Purification by silica gel column chromatography (eluent: AcOEt/*n*-hexane = 1/3) gave **4a** as a colorless solid (358 mg, 1.33 mmol, 97 %); colorless needles (MeOH/CH_2_Cl_2_); mp 168.5–170.0 °C; ^1^H NMR (400 MHz, CD_3_OD) *δ* (ppm) 4.15 (4H, s), 6.62 (4H, d, *J* = 8.8 Hz), 6.95 (4H, d, *J* = 8.8 Hz); ^13^C NMR (100 MHz, CD_3_OD) *δ* (ppm) 44.3, 116.0, 129.8, 130.8, 135.5, 157.7; MS (EI) *m/z* = 270 (M^+^, 100%); HRMS Calcd for C_16_H_14_O_2_S: 270.0715, Found: 270.0707; Anal. Calcd for C_16_H_14_O_2_S 0.4 H_2_O: C, 69.23; H, 5.38, Found: C, 69.04; H, 5.66.

*3,4-Bis-(4-methoxyphenyl)-2,5-dihydrothiophene-1-oxide* (**9**). To a solution of **8** (406 mg, 1.36 mmol) in dry CH_2_Cl_2_ (15 mL) was portionwise added *m*-CPBA (65%, 349 mg, 1.41 mmol) at 0 °C. The reaction mixture was stirred at room temperature for 25 h and then quenched with saturated Na_2_SO_3_ aqueous solution. The mixture was extracted with AcOEt, washed with saturated NaHCO_3_ aqueous solution and brine, dried over MgSO_4_, and concentrated. Purification by silica gel column chromatography (eluent: AcOEt/*n*-hexane = 1/1) gave **9** as a colorless solid (302 mg, 0.96 mmol, 70%); colorless needles (CHCl_3_/*n*-hexane); mp 181.0–182.0 °C; ^1^H NMR (400 MHz, CDCl_3_) *δ* (ppm) 3.78 (6H, s), 4.00 (2H, d, *J* = 16.6 Hz), 4.34 (2H, d, *J* = 16.6 Hz), 6.77 (4H, d, *J* = 8.8 Hz), 7.10 (4H, d, *J* = 8.8 Hz); ^13^C NMR (100 MHz, CDCl_3_) *δ* (ppm) 55.2, 64.3, 113.9, 128.1, 129.9, 130.1, 159.1; MS (EI) *m/z* = 296 (M^+^, 100%); HRMS Calcd for C_18_H_18_O_3_S: 314.0977, Found: 314.0974; Anal. Calcd for C_18_H_18_O_3_S: C, 68.76; H, 5.77, Found: C, 68.42; H, 5.88.

*3,4-Bis-(4-hydroxyphenyl)-2,5-dihydrothiophene-1-oxide* (**4b**). To a solution of **4a** (402 mg, 1.49 mmol) in dry CH_2_Cl_2_ (60 mL) was portion-wise added *m*-CPBA (65%, 368 mg, 1.49 mmol) at 0 °C. The reaction mixture was stirred at room temperature for 20 h and then quenched with saturated Na_2_SO_3_ aqueous solution. The mixture was extracted with AcOEt, washed with saturated NaHCO_3_ aqueous solution and brine, dried over MgSO_4_, and concentrated. Purification by silica gel column chromatography (eluent: AcOEt/*n*-hexane = 1/1) afforded **4b** as a colorless solid (311 mg, 1.09 mmol, 73 %); colorless needles (MeOH/CH_2_Cl_2_); mp 219.0–220.0 °C; ^1^H NMR (400 MHz, CD_3_OD) *δ* (ppm) 3.94 (2H, d, *J* = 17.1 Hz), 4.49 (2H, d, *J* = 17.1 Hz), 6.67 (4H, d, *J* = 8.8 Hz), 7.05 (4H, d, *J* = 8.8 Hz); ^13^C NMR (100 MHz, CD_3_OD) *δ* (ppm) 64.9, 116.3, 128.4, 131.0, 131.1, 158.3; MS (EI) *m/z* = 286 (M^+^), 268 (100%); HRMS Calcd for C_16_H_14_O_3_S: 286.0664, Found: 286.0665; Anal. Calcd for C_16_H_14_O_3_S 0.1 H_2_O: C, 66.69; H, 4.97, Found: C, 66.47; H, 4.72.

*3,4-Bis-(4-methoxyphenyl)-2,5-dihydrothiophene-1,1-dioxide* (**10**). To a solution of **8** (601 mg, 2.02 mmol) in dry CH_2_Cl_2_ (30 mL) was portionwise added *m*-CPBA (65%, 1.11 g, 4.51 mmol) at 0 °C. The reaction mixture was stirred at room temperature for 3.5 h and then quenched with saturated Na_2_SO_3_ aqueous solution. The mixture was extracted with AcOEt, washed with saturated NaHCO_3_ aqueous solution and brine, dried over MgSO_4_, and concentrated. The residue was purified by recrystallization with CHCl_3_/*n*-hexane to give **10** (377 mg, 1.14 mmol, 57 %) as a colorless needles; mp 141.0–141.5 °C; ^1^H NMR (400 MHz, CDCl_3_) *δ* (ppm) 3.78 (6H, s), 4.26 (4H, s), 6.77 (4H, d, *J* = 9.3 Hz), 7.03 (4H, d, *J* = 9.3 Hz); ^13^C NMR (100 MHz, CDCl_3_) *δ* (ppm) 55.2, 61.0, 114.0, 127.2, 129.2, 129.7, 159.4; MS (EI) *m/z* = 330 (M^+^), 266 (100%); Anal. Calcd for C_18_H_18_O_4_S: C, 65.43; H, 5.49, Found: C, 65.15; H, 5.55.

*3,4-Bis-(4-hydroxyphenyl)-2,5-dihydrothiophene-1,1-dioxide* (**4c**). To a solution of **10** (203 mg, 0.61 mmol) in dry CH_2_Cl_2_ (10 mL) was added 1 M of a solution of BBr_3_ in CH_2_Cl_2_ (1.4 mL, 1.4 mmol) at −80 °C. The reaction mixture was stirred at room temperature for 24 h, poured onto ice, and then extracted with AcOEt. The organic layer was washed with brine, dried over MgSO_4_, and concentrated. Purification by silica gel column chromatography (eluent: AcOEt/*n*-hexane = 1/2) afforded **4c** as a colorless solid (185 mg, 0.61 mmol, quant); colorless needles (AcOEt/*n*-hexane); mp 180.5–181.0°C; ^1^H NMR (400 MHz, CD_3_OD) *δ* (ppm) 4.27 (4H, s), 6.64 (4H, d, *J* = 8.8 Hz), 6.99 (4H, d, *J* = 8.8 Hz); ^13^C NMR (100 MHz, CD_3_OD) *δ* (ppm) 61.7, 116.2, 127.9, 130.5, 131.0, 158.6; MS (EI) *m/z* = 302 (M^+^), 238 (100%); HRMS Calcd for C_16_H_14_O_4_S: 302.0613, Found: 302.0618; Anal. Calcd for C_16_H_14_O_4_S: C, 63.56; H, 4.67, Found: C, 63.29; H, 4.75.

*4-{4-[4-(3-dimethylaminopropoxy)phenyl]-2,5-dihydrothiophen-3-yl}phenol* (**4d**). To a solution of **4a** (100 mg, 0.37 mmol) in dry THF (5 mL), 3-dimethylamino-1-propanol (48 μL, 0.41 mmol), triphenylphosphine (97 mg, 0.37 mmol) was added a solution of diethyl azodicarboxylate in toluene (40%, 161 μL, 0.37 mmol) at room temperature, and the mixture was stirred for 22 h. The mixture was concentrated and purified by silica gel column chromatography (eluent: MeOH/CHCl_3_ = 1/5) gave **4d** as a colorless solid (49 mg, 37%); colorless needles (AcOEt); mp 177.0–178.5°C; ^1^H NMR (400 MHz, DMSO-*d_6_*) *δ* (ppm) 1.79 (2H, quint, *J* = 6.5 Hz), 2.11 (6H, s), 2.30 (2H, t, *J* = 7.2 Hz), 3.92 (2H, t, *J* = 6.0 Hz), 4.14 (4H, s), 6.60 (2H, d, *J* = 8.2 Hz), 6.76 (2H, d, *J* = 8.7 Hz), 6.93 (2H, d, *J* = 7.7 Hz), 7.03 (2H, d, *J* = 8.2 Hz), 9.45 (1H, s); ^13^C NMR (100 MHz, DMSO-*d_6_*) *δ* (ppm) 26.9, 43.0, 45.2, 55.6, 65.6, 114.1, 115.1, 126.8, 128.2, 128.9, 129.5, 133.1, 134.1, 157.2, 157.6; MS (EI) *m/z* 355 (M^+^), 58 (100%); HRMS Calcd for C_21_H_25_NO_2_S: 355.1606, Found: 355.1607.

*3-Chloro-1-(4-methoxyphenyl)-propan-1-one* (**12**). To a soltion of anisol (8.5 mL, 78.2 mmol) and AlCl_3_ (11 g, 82.5 mmol) in dry 1,2-dichloroethane (30 mL) was added 3-chloropropionyl chloride **11** (5.5 mL, 57.61 mmol) at 0 °C, and the mixture was stirred at room temperature for 2 h. The mixture was poured onto ice and extracted with AcOEt. The organic layer was washed with brine, dried over MgSO_4_, and concentrated. The residue was purified by recrystallization from CH_2_Cl_2_/*n*-hexane to give **12** (9.3 g, 46.7 mmol, 81%) as a colorless solid; ^1^H NMR (400 MHz, CDCl_3_) *δ* (ppm) 3.40 (2H, t, *J* = 6.8 Hz), 3.87 (3H, s), 3.91 (2H, t, *J* = 6.8 Hz), 6.95 (2H, d, *J* = 8.8 Hz), 7.94 (2H, d, *J* = 8.8 Hz); ^13^C NMR (100 MHz, CDCl_3_) *δ* (ppm) 38.9, 40.9, 55.5, 113.9, 129.5, 130.3, 163.8, 195.2; MS (EI) *m/z* = 198 (M^+^), 135 (100%).

*7-Methoxy-1-[3-(4-methoxy-phenyl)-3-oxo-propylsulfanyl]-4-methyl-hepta-4,6-dien-3-one* (**13**). To a stirred solution of **12** (7.01 g, 35.4 mmol) in acetone (100 mL) was added a solution of Na_2_S 9H_2_O (4.12 g, 17.2 mmol) in H_2_O (12 mL) at 0 °C. After completion of the addition, the mixture was warmed to room temperature and stirred for 21 h. The solvents were removed, and the residue was recrystallized from MeOH to give **13** (6.79 g, 17.4 mmol, 98 %) as a colorless solid; ^1^H NMR (400 MHz, CDCl_3_) *δ* (ppm) 3.0 (4H, t, *J* = 7.3 Hz), 3.2 (4H, t, *J* = 7.3 Hz), 3.9 (6H, s), 6.9 (4H, d, *J* = 8.8 Hz), 7.9 (4H, d, *J* = 8.8 Hz); ^13^C NMR (100 MHz, CDCl_3_) *δ* (ppm) 26.7, 38.5, 55.5, 113.8, 129.7, 130.3, 163.6, 196.8. MS (EI) *m/z* = 358 (M^+^), 135 (100%).

*4,5-Bis (4-methoxyphenyl)-2,3,6,7-tetrahydrothiepine* (**14**). To a suspension of Zn powder (4.39 g, 67.1 mmol) in THF (90 mL) was added TiCl_4_ (3.8 mL, 34.6 mmol) at −30 °C, and the mixture was refluxed for 2.5 h. A solution of **13** (3.01 g, 8.4 mmol) in THF (120 mL) was slowly added to the reaction mixture at 0 °C. The reaction mixture was refluxed for 18 h and then quenched with 10% K_2_CO_3_ aqueous solution. The insoluble materials were filtered through Celite, and the filtrate was extracted with AcOEt. The organic layer was washed with brine, dried over MgSO_4_, and concentrated. The residue was purified by column chromatography on silica gel with *n*-hexane to give **14** (2.72 g, 8.34 mmol, 99%) as a colorless solid; ^1^H NMR (400 MHz, CDCl_3_) *δ* (ppm) 2.8 (4H, t, *J* = 5.3 Hz), 3.13 (4H, t, *J* = 5.3 Hz), 3.72 (6H, s), 6.65 (4H, d, *J* = 8.8 Hz), 6.88 (4H, d, *J* = 8.8 Hz); ^13^C NMR (100 MHz, CDCl_3_) *δ* (ppm) 27.3, 40.0, 55.1, 113.2, 130.0, 136.8, 139.2, 157.5; MS (EI) *m/z* = 326 (M^+^, 100%).

*3,4-Bis (4-hydroxyphenyl)-2,3,6,7-tetrahydrothiepine* (**5a**). To a solution of **14** (1.01 g, 3.08 mmol) in dry CH_2_Cl_2_ (15 mL) was added 1M of a solution of BBr_3_ in CH_2_Cl_2_ (7.6 mL, 7.6 mmol) at −80 °C, and the mixture was stirred at room temperature for 16 h. The mixture was poured onto ice and extracted with AcOEt. The organic layer was washed with brine, dried over MgSO_4_, and concentrated. The residue was purified by column chromatography on silica gel with 1:2 AcOEt:*n*-hexane to give **5a** (413 mg, 2.02 mmol, 66 %) as a colorless solid; colorless needles (MeOH/CH_2_Cl_2_); mp 165.5–166.0 °C; ^1^H NMR (400 MHz, CD_3_OD) *δ* (ppm) 2.73 (4H, t, *J* = 5.4 Hz), 3.09 (4H, t, *J* = 5.4 Hz), 6.51 (4H, d, *J* = 8.8 Hz), 6.78 (4H, d, *J* = 8.8 Hz); ^13^C NMR (100 MHz, CD_3_OD) *δ* (ppm) 28.1, 41.0, 115.5, 131.2, 137.4, 140.5, 156.3; MS (EI) *m/z* = 298 (M^+^, 100%); HRMS Calcd for C_18_H_18_O_2_S: 298.1028, Found: 298.1034; Anal. Calcd for C_18_H_18_O_2_S: C, 72.45; H, 6.08, Found: C, 72.27; H, 6.15.

*4,5-Bis (4-hydroxyphenyl)-2,3,6,7-tetrahydrothiepine-1-oxide* (**5b**). To a solution of **5a** (402 mg, 1.35 mmol) in CH_2_Cl_2_ (60 mL) was portionwise added *m*-CPBA (65%, 399 mg, 1.62 mmol) at 0 °C. The reaction mixture was stirred at room temperature for 9 h and then quenched with saturated Na_2_SO_3_ aqueous solution. The mixture was extracted with AcOEt, washed with saturated NaHCO_3_ aqueous solution and brine, dried over MgSO_4_, and concentrated. The residue was purified by column chromatography on silica gel with 3:1 AcOEt:*n*-hexane to give **5b** (127 mg, 0.40 mmol, 30 %) as a colorless solid; colorless needles (MeOH/CH_2_Cl_2_); mp 190–191.5 °C; ^1^H NMR (400 MHz, CD_3_OD) *δ* (ppm) 2.51 (2H, dd, *J* = 15.8 Hz), 3.00 (2H, dd, *J* = 14.2 Hz), 3.10 (2H, t, *J* = 14.2 Hz), 3.65 (2H, dd, *J* = 15.6 Hz), 6.54 (4H, d, *J* = 8.3 Hz), 6.83 (4H, d, *J* = 8.3 Hz); ^13^C NMR (100 MHz, CD_3_OD) *δ* (ppm) 26.0, 46.8, 115.9, 131.4, 136.2, 140.4, 157.1; MS (EI) *m/z* = 314 (M^+^), 251 (100%); HRMS Calcd for C_18_H_18_O_3_S: 314.0977, Found: 314.0976.

*4,5-Bis (4-methoxyphenyl)-2,3,6,7-tetrahydrothiepine-1,1-dioxide* (**15**). To a solution of **14** (1.00 mg, 3.07 mmol) in dry CH_2_Cl_2_ (20 mL) was portion-wise added *m*-CPBA (65%, 1.75 g, 7.11 mmol) at 0 °C. The reaction mixture was stirred at room temperature for 22 h and then quenched with saturated Na_2_SO_3_ aqueous solution. The mixture was extracted with AcOEt, washed with saturated NaHCO_3_ aqueous solution and brine, dried over MgSO_4_, and concentrated. The residue was purified by column chromatography on silica gel with 1:3 AcOEt:*n*-hexane to give **15** (1.05 g, 2.93 mmol, 96 %) as colorless solid; ^1^H NMR (400 MHz, CDCl_3_) *δ* (ppm) 3.02 (4H, t, *J* = 5.1 Hz), 3.17 (4H, t, *J* = 5.1 Hz), 3.74 (6H, s), 6.68 (4H, d, *J* = 8.8 Hz), 6.88 (4H, d, *J* = 8.8 Hz); ^13^C NMR (100 MHz, CDCl_3_) *δ* (ppm) 30.4, 54.0, 55.3, 113.7, 130.1, 135.1, 138.6, 158.3. MS (EI) *m/z* = 358 (M^+^), 292 (100%).

*4,5-Bis (4-hydroxyphenyl)-2,3,6,7-tetrahydrothiepine-1,1-dioxide* (**5c**). To a solution of **15** (403 mg, 1.12 mmol) in dry CH_2_Cl_2_ (7 mL) was added 1M of a solution of BBr_3_ in CH_2_Cl_2_ (2.8 mL, 2.8 mmol) at −80 °C. The reaction mixture was stirred at room temperature for 31 h, poured onto ice, and then extracted with AcOEt. The organic layer was washed with brine, dried over MgSO_4_, and concentrated. The residue was purified by column chromatography on silica gel with 1 : 2 AcOEt : *n*-hexane to give **5c** (195 mg, 0.20 mmol, 53%) as a colorless solid; colorless needles (AcOEt/*n*-hexane); mp 128.5–130.0 °C; ^1^H NMR (400 MHz, CD_3_OD) *δ* (ppm) 2.97 (4H, t, *J* = 5.3 Hz), 3.21 (4H, t, *J* = 5.3 Hz), 6.54 (4H, d, *J* = 8.8 Hz), 6.85 (4H, d, *J* = 8.8 Hz); ^13^C NMR (100 MHz, CD_3_OD) *δ* (ppm) 31.2, 54.5, 115.7, 131.4, 135.8, 139.8, 157.0; MS (EI) *m/z* = 330 (M^+^), 264 (100%); HRMS Calcd for C_18_H_18_O_4_S: 330.0926, Found: 330.0920; Anal. Calcd for C_18_H_18_O_4_S: C, 65.43; H, 5.49, Found: C, 65.40; H, 5.59.

*4-{5-[4-(3-Dimethylaminopropoxy)phenyl]-2,3,6,7-tetrahydrothiepin-4-yl}phenol* (**5d**). To a solution of **5a** (200 mg, 0.67 mmol) in dry THF (10 mL), 3-dimethylamino-1-propanol (87 μL, 0.74 mmol), triphenylphosphine (176 mg, 0.67 mmol) was added a solution of diethyl azodicarboxylate in toluene (40%, 292 μL, 0.67 mmol) at room temperature, and the mixture was stirred for 22 h. After the solvent was removed, the residue was purified by column chromatography on silica gel with 1:5 MeOH:CHCl_3_ to give **5d** (71 mg, 28%) as a colorless solid; Colorless needles (CH_2_Cl_2_/MeOH); mp 176.0–177.0 °C; ^1^H NMR (400 MHz, CDCl_3_) *δ* (ppm) 1.88 (2H, quint, *J* = 6.9 Hz), 2.25 (6H, s), 2.43 (2H, t, *J* = 7.7 Hz), 2.78 (4H, t, *J* = 4.8 Hz), 3.12 (4H, t, *J* = 5.1 Hz), 3.91 (2H, t, *J* = 6.3 Hz), 6.53 (2H, d, *J* = 8.7 Hz), 6.61 (2H, d, *J* = 8.7 Hz), 6.79 (2H, d, *J* = 8.7 Hz), 6.83 (2H, d, *J* = 8.7 Hz); ^13^C NMR (100 MHz, CDCl_3_) *δ* (ppm) 27.1, 27.3, 39.9, 45.2, 56.3, 66.0, 113.8, 115.0, 130.1, 130.2, 136.3, 137.0, 139.1, 139.5, 154.4, 156.7; MS (EI) *m/z* 383 (M^+^), 58 (100%); HRMS Calcd for C_23_H_29_NO_2_S: 383.1919, Found: 383.1931.

### 3.3. Competitive Binding Assay Using Human ER

The ligand binding activity of human estrogen receptor α (ER α) was determined by a nitrocellulose filter binding assay method. ER α was diluted with a binding assay buffer (20 mM Tris-HCl pH 8.0, 0.3 M NaCl, 1 mM EDTA pH 8.0, 10 mM 2-mercaptoethanol, 0.2 mM phenylmethylsulfonyl floride) and incubated with 4 nM [6, 7-^3^H]-17β-estradiol in the presence or absence of an unlabeled competitor at 4 °C for 18 h. The incubation mixture was absorbed by suction onto a nitrocellulose membrane that had been soaked in binding assay buffer. The membrane was washed two times with buffer (20 mM Tris-HCl pH 8.0, 0.15 M NaCl) and with 25% ethanol in distilled water. Radioactivity that remained in the membrane was measured in Atomlight by using a liquid scintillation counter.

### 3.4. MCF-7 Cell Proliferation Assay

The human breast adenocarcinoma cell line MCF-7 was routinely cultivated in DMEM supplemented with 10% FBS and 100 IU/mL penicillin and 100 mg/mL streptomycin at 37 °C in a 5% CO_2_ humidified incubator. Before an assay, MCF-7 cells were switched to DMEM (low glucose phenol red-free supplemented with 5% FBS, and 100 IU/mL penicillin and 100 mg/mL streptomycin). Cells were trypsinized from the maintenance dish with phenol red-free trypsin-EDTA and seeded in a 96-well plate at a density of 2000 cells per final volume of 100 μL DMEM supplemented with 5% stripped FBS and 100 IU/mL penicillin and 100 mg/mL streptomycin. After 24 h, the medium was removed and 90 μL of fresh medium and 10 μL of drug solution, supplemented with serial dilutions of test compounds or DMSO as dilute control in the presence or absence of 0.1 nM E2, were added to triplicate microcultures. Cells were incubated for four days. At the end of the incubation time, number of cells was counted by using the WST-8, which was added to microcultures 10 μL each, and they were incubated for 2–4 h. The absorbance at 450 nm was measured. This parameter relates to and number of living cells in the culture.

### 3.5. Docking Simulation Study

Three-dimensional (3D) structures of protein-ligand complexes were predicted using the Discovery Studio 2018/CDOCKER software (BIOVIA) with default settings. The 3D structures of ER used in this study were retrieved from the RCSB Protein Data Bank (PDB ID: 1ERR). Missing hydrogen atoms in the crystal structure were computationally added and the center of the active site was defined as the center of raloxifen in 1ERR. The conformations of **4d** and **5d** were optimized using the CHARMm force field, and the docking simulations of **4d** and **5d** with 1ERR were performed using the CDOCKER protocol. The docking poses for **4d** and **5d** with the highest CDOCKER ENERGY were selected for the discussions of 10 docking modes, respectively.

## 4. Conclusions

Novel sulfur-containing ERα modulators (compounds **4** and **5**) as potential breast cancer therapeutic drug candidates were designed and synthesized based upon the hydrophobic feedback approach for the simple bisphenols **1**–**3** developed in our previous studies. Tetrahydrothiepine derivatives (compounds **5a**–**5d**) showed higher binding affinity toward ERα than the corresponding dihydrothiophene derivatives (compounds **4a**–**4d**) because of the hydrophobicity of the sulfur-containing ring structure. Although the bisphenol derivatives **4a**, **4c**, **5a**, and **5c** showed estrogenic activity toward the MCF-7 breast cancer cell line, the corresponding alkylamino derivatives (compounds **4d** and **5d**) acted as ER antagonists. In particular, compound **5d** showed the most potent antiestrogenic activity among the tested compounds with an IC_50_ value of 0.09 μM. A sulfur-containing structure might be a promising scaffold for antiestrogen discovery, owing to ease of synthesis, binding modes toward the ERα LBD, and availability of various substituted derivatives.

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
