# Peer review of "Design and Synthesis of Novel Breast Cancer Therapeutic Drug Candidates Based upon the Hydrophobic Feedback Approach of Antiestrogens"

_molecules, 2019, doi:10.3390/molecules24213966_

Round 1

Reviewer 1 Report

This paper deals with the synthesis of new sulfur-containing analogues of tamoxifen and their performance as ERa modulators and anti-breast cancer drugs.

The authors have published extensive previous work in this field with different tamoxifen-inspired compounds. The synthesis of the new compounds is well described, and the biological results are promising, specially those related with compound 5a. Therefore, the paper deserves to be published in Molecules

However, for the complete characterization of relevant new compounds, the melting point of solid compounds is mandatory, especially for final products 4 and 5. Thus melting point of compounds 5a-c should be provided.

On the other hand, a spelling mistake must be corrected in the captions of Scheme 1 and 2 (3-dimethylamino-1-propanol).

Reviewer 2 Report

The paper by K.Ohta et al. describes the design and synthesis of novel sulfur containing breast cancer therapeutic drug candidates. The dihydrothiophene and tetrahydrothiepine derivatives showed the highest binding affinity toward ERα. 

From synthetic point of view, the paper is well organized, the experimental details are well described, all new compounds are characterized by C, H, and N microanalyses and 1H and 13C NMR spectra. The paper can be published as such.

However, I can not say anything about biological evaluation, since it is not my area.

Reviewer 3 Report

Referee for molecules-617139

Novel ERα modulators have been designed and synthesized as breast cancer agents. Structural characterization of the compounds has been done by 1HNMR and 13CNMR. Their Purity of final compounds has been controlled by HRMS and elemental analyses. Binding affinity has been determined by radioligand displacement studies. Their antiproliferative properties have been evaluated in MTT assays in MCF-7 cell. Docking studies intent explaining the interactions of the hydrophobic moiety of compounds with the hydrophobic pocket.

One of the compounds, 5d, showed activity inhibiting MCF-7 14 cell proliferation withIC50 of 0.09 micromolar.

This manuscript can be considered for publishing in Molecules after some revisions.

Referee’s suggestions:

Introduction

Lines 31-37. The following sentences need to be clarified: “The dimethylaminoethyl chain of tamoxifen is attached to a benzene ring where it plays a strategic role in the expression of antiestrogenic activity”….” Although the triphenylethylene moiety is a promising structure for the development of an ER ligand, it exhibits geometric isomers E and Z that are caused by the key alkylamino chain and the asymmetric hydrophobic part.”

Line 40. “…. structures, have been developed and used as ER 40 antagonists”. What do the authors mean by use as ER40 antagonists? Are they used in clinical trials? Are they commercialized?

General. In the introduction, the authors do not explain the rational for the design of the sulfide, sulfone and sulfoxide. This is quite important since there is no such functional group in the previous reference ER modulators showed in Figure 1.

Synthesis

The synthesis is well described.

Binding assays

The binding assays section should be improved. What about the Ki or % of radioligand displacement?

Figure 2: Are EC50 and IC50 expressed in  micromolar, nanomolar? Error data?

Since the study is based on an hydrophobic approach, cLogP should have been determined experimentally but not as predictive data.

Docking study

Energy of interaction between the ligand and residues in the binding site?

Conformational cost to assume the conformation in the final complex?

Experimental

Melting points of 7,8,12,13,5a … are missing.

It will have been interesting having the NMR signal attributions.

Author Response

Responses to reviewer comments and suggestions (reviewer 3).

Novel ERα modulators have been designed and synthesized as breast cancer agents. Structural characterization of the compounds has been done by 1HNMR and 13CNMR. Their Purity of final compounds has been controlled by HRMS and elemental analyses. Binding affinity has been determined by radioligand displacement studies. Their antiproliferative properties have been evaluated in MTT assays in MCF-7 cell. Docking studies intent explaining the interactions of the hydrophobic moiety of compounds with the hydrophobic pocket. One of the compounds, 5d, showed activity inhibiting MCF-7 14 cell proliferation withIC50 of 0.09 micromolar. This manuscript can be considered for publishing in Molecules after some revisions.

Referee’s suggestions:

Introduction

Q1.

Lines 31-37. The following sentences need to be clarified: “The dimethylaminoethyl chain of tamoxifen is attached to a benzene ring where it plays a strategic role in the expression of antiestrogenic activity”….” Although the triphenylethylene moiety is a promising structure for the development of an ER ligand, it exhibits geometric isomers E and Z that are caused by the key alkylamino chain and the asymmetric hydrophobic part.”

We added the explanation that the isomerization between the E and Z isomers often become a problem for the development of ER ligands.

Q2.

Line 40. “…. structures, have been developed and used as ER antagonists”. What do the authors mean by use as ER antagonists? Are they used in clinical trials? Are they commercialized?

We changed the sentence as follows: “have been developed as ER antagonists”.

Q3.

General. In the introduction, the authors do not explain the rational for the design of the sulfide, sulfone and sulfoxide. This is quite important since there is no such functional group in the previous reference ER modulators showed in Figure 1.

We added the following sentence as the explanation for the design of sulfide, sulfone, and sulfoxide derivatives: “We were interested in the effects of ring size and sulfur atom in the hydrophobic structure on ER activity. Furthermore, to understand the effects of sulfur, sulfone, and sulfoxide on the biological activity”

Synthesis

The synthesis is well described.

Thank you very much for your kind comments.

Binding assays

Q4.

The binding assays section should be improved. What about the Ki or % of radioligand displacement?

IC50 value of the E2 was shown in the legend of Figure 2. Since the relative binding affinity (RBA) for the labeled E2 mean %, a unit (%) for RBA was shown in Table 1.

Q5.

Figure 2: Are EC50 and IC50 expressed in  micromolar, nanomolar? Error data?

I think the reviewer mix up Figure 2 and Table 1. Figure 2 is the data of binding assay and is not related with EC50 and IC50 values, which were estimated from cell proliferation assay. Therefore, units of the horizontal axis of Figure 2 mean “times”, not “molar”. Units for EC50 and IC50 had been shown in Table1 of our submitted manuscript. Unfortunately, these units were out of the titles in Table 1. To take back the units, we changed the font size of EC50 and IC50 in Table1. Since the binding assays were performed in duplicate (n=2), we cannot obtain the standard deviation (SD) values.

Q6.

Since the study is based on an hydrophobic approach, cLogP should have been determined experimentally but not as predictive data.

As pointed out by the reviewer, our research is based upon the hydrophobic approach. Main discussion of the manuscript do not include quantitative structure-activity relationship study between hydrophobicity and biological activities. We use the parameter of cLogP only as a guide for the hydrophobicity, not for the delicate discussion. Therefore, cLogP is enough for this study.

Docking study

Q7.

Energy of interaction between the ligand and residues in the binding site?

We have already described the docking score of 4d and 5d in the first submitted manuscript. Please confirm the following description at the line 155-156: “Docking scores (CDOCKER interaction energy) of compounds 4d and 5d toward the ERa LBD were 51.1 and 54.0, respectively” Discovery Studio 2018 cannot estimate the partial docking energy between the ligand with the specific amino acid residues.

Q8.

Conformational cost to assume the conformation in the final complex?

Does conformational cost mean the conformational energy for 4d and 5d? In the estimated complex structure, the conformational energy of 4d and 5d are 13.8 and 4.7, respectively. It will be reasonable that compound 4d is energetically more stable than 5d, because compound 4d has a 5-membered ring system.

Experimental

Q9.

Melting points of 7,8,12,13,5a … are missing.

1H NMR and MS data are enough for the determination of the known compounds. Therefore, the crystal data including mp of the used compounds for biological assays are shown in experimental section.

Q10.

It will have been interesting having the NMR signal attributions.

The target compounds have simple and symmetric chemical structures, In this case, we think that NMR signal attributions for these compounds are not useful and interesting information.

Reviewer 4 Report

The manuscript by Ohta et al. describes synthesis and biological evaluation of eight novel dihydrothiophene and tetrahydrothiepine derivatives. Obtained compounds were designed as estrogen receptor antagonists. However only two compounds bearing dimethylaminopropyl group displayed desired activity. The manuscript is clearly written. Although the synthesis is straightforward it nevertheless contains interesting results from biological characterisation of prepared molecules. However there are some minor issues I would like to point out:

1. line 120 - Please provide explanation why compounds 4b and 5b were not evaluated in the cell assay (poor binding?)

2. line 125 - Unless authors define what they exactly mean by activity, the sentence is not consequential. Differences in Emax can be attributed to efficacy of ligands which is a measure of activity as well. Please consider rephrasing this sentence.

3. line 135 - Correlation of antiestrogenic activity to the binding affinity does not seem logical. Compound 4d has better affinity than 5a, yet it displayed estrogenic activity rather than antiestrogenic. It seems that the presence of dimethylaminopropyl group is responsible for antiestrogenic activity, not increased binding. This issue can be perhaps elaborated more in the docking section. Are there any interactions with the dimethylpropyl chain that would explain observed activity? Also Figure 3A and 3B would benefit from being prepared again (eg. hide non-polar hydrogens, show secondary structure). Please address this concern.

4. line 44 - Please provide explanation for OVX and ORX abbrevations.

5. line 49 - bicyclo[2.2.2]octene

6. line 61 - 2-bromo-4'-methoxyacetophenone or ω-bromo-4-methoxyacetophenone

7. line 73 - In the scheme 8 -> 10 -> 4c there is a wrong order of reagents (should be d and then c rather than c and then d).

8. line 76 - propanol

9. line 78 - acylation of anisole with acyl chloride 11 rather than the other way around

10. line 91 - propanol

11. line 111 - Table 1. please add units in the header for EC50 and IC50.

12. line 162 - compounds

13. line 213, 224, 235, 304, 314 - mCPBA is a solid substance. How can this be added dropwise?

14. line 270 - Please specify which solvent was used for recrystallisation.

15. line 347 and 351 - nitrocellulose

16. line 368 - number?

Issues listed above do not detract from the value of this manuscript and after being addressed I recommend the manuscript to be published.

Round 2

Reviewer 3 Report

The reviewer suggests that the manuscript can be published in Molecules.